# Phospho-RNA-Seq Highlights Specific Small RNA Profiles in Plasma Extracellular Vesicles

**DOI:** 10.3390/ijms241411653

**Published:** 2023-07-19

**Authors:** Maria Solaguren-Beascoa, Ana Gámez-Valero, Georgia Escaramís, Marina Herrero-Lorenzo, Ana M. Ortiz, Carla Minguet, Ricardo Gonzalo, Maria Isabel Bravo, Montserrat Costa, Eulàlia Martí

**Affiliations:** 1Department de Biomedicina, Facultat de Medicina i Ciències de la Salut, Institut de Neurociències, Universitat de Barcelona, C/Casanova 143, 08036 Barcelona, Spain; m.solaguren-beascoa@ub.edu (M.S.-B.); a.gamez@ub.edu (A.G.-V.); gescaramis@ub.edu (G.E.); marina.herrero@ub.edu (M.H.-L.); 2Centro de Investigación Biomédica en Red de Epidemiología y Salud Pública (CIBERESP), Ministerio de Ciencia Innovación y Universidades, 28029 Madrid, Spain; 3Grifols Scientific Innovation Office, 08022 Barcelona, Spain; aortizfe@hotmail.com (A.M.O.); carla.minguet@grifols.com (C.M.); ricardo.gonzalo@grifols.com (R.G.); isabel.bravo@grifols.com (M.I.B.); montse.costa@grifols.com (M.C.)

**Keywords:** biomarkers, extracellular vesicles, plasma, T4-PNK, small RNA

## Abstract

Small RNAs (sRNAs) are bioactive molecules that can be detected in biofluids, reflecting physiological and pathological states. In plasma, sRNAs are found within extracellular vesicles (EVs) and in extravesicular compartments, offering potential sources of highly sensitive biomarkers. Deep sequencing strategies to profile sRNAs favor the detection of microRNAs (miRNAs), the best-known class of sRNAs. Phospho-RNA-seq, through the enzymatic treatment of sRNAs with T4 polynucleotide kinase (T4-PNK), has been recently developed to increase the detection of thousands of previously inaccessible RNAs. In this study, we investigated the value of phospho-RNA-seq on both the EVs and extravesicular plasma subfractions. Phospho-RNA-seq increased the proportion of sRNAs used for alignment and highlighted the diversity of the sRNA transcriptome. Unsupervised clustering analysis using sRNA counts matrices correctly classified the EVs and extravesicular samples only in the T4-PNK treated samples, indicating that phospho-RNA-seq stresses the features of sRNAs in each plasma subfraction. Furthermore, T4-PNK treatment emphasized specific miRNA variants differing in the 5′-end (5′-isomiRs) and certain types of tRNA fragments in each plasma fraction. Phospho-RNA-seq increased the number of tissue-specific messenger RNA (mRNA) fragments in the EVs compared with the extravesicular fraction, suggesting that phospho-RNA-seq favors the discovery of tissue-specific sRNAs in EVs. Overall, the present data emphasizes the value of phospho-RNA-seq in uncovering RNA-based biomarkers in EVs.

## 1. Introduction

Small RNAs (sRNA) are regulatory species involved in the targeted modulation of gene expression at different levels. The highly regulated expression of the different types of sRNAs is key for the control of cell physiology, and the identification of meaningful species is an active field of research.

The expression patterns of sRNAs are highly dynamic, reflecting the cell physiological and pathological states. Thus, the detection of extracellular sRNAs in biofluids, such as blood, has opened new avenues in the search for disease and biological biomarkers [1,2,3]. In plasma, sRNA can be found freely circulating and/or enclosed in extracellular vesicles (EVs), which are naturally released by all types of cells mediating cell-to-cell communication [4,5,6,7]. EV sRNAs are protected from the milieu RNAses [8,9], and their characterization represents a readout of the cell/tissue status from where they originate. Given the fine tissue and cell status specificity of the transcriptome, EV sRNAs are highlighted as a significant source of biomarkers [10,11]. The mechanisms underlying RNA loading involve specific sequence features of RNAs along with RNA-binding proteins (RBP), whose competitive interplay is emerging as a crucial determinant in diversifying the enrichment of the selected transcripts into the EVs [12]. These include members of the hnRNP family, YBX1, HuR, and AGO2, among others, and the related mechanism have been widely reviewed by the authors of [12].

To date, the characterization of sRNA content within EVs has been largely focused on microRNAs (miRNAs), the best-known class of sRNAs. However, miRNAs represent a minor portion of the EV transcriptome. Several studies have revealed that other types of sRNAs, including fragments of messenger RNA (mRNAs), transfer RNA (tRNA), ribosomal RNA (rRNA), and Y RNA (yRNA), are highly abundant in EVs [13,14,15,16,17], and can be similarly exploited for biomarker discovery in human diseases, such as cancer [18,19,20] and neurological disorders [21]. Furthermore, an important part of the circulating RNA is not associated with EVs [22,23], and its potential as a source of biomarkers has not been explored.

In recent years, advances in sRNA sequencing strategies have allowed us to analyze and decipher the complex composition of the human transcriptome across a wide range of contexts [24,25]. Standard ligation-based sRNA-sequencing methods have been designed to optimally capture miRNAs which, by virtue of being products of RNase III class enzymes (e.g., Dicer), consistently present 5′monophosphate (5′P) and 3′hydroxyl (3′OH) ends [26,27]. However, the detection of RNA fragments generated by ribonucleases that produce alternative 5′ and 3′ ends is impaired as these ends hinder efficient adapter ligation. This problem can be overcome through the treatment of RNAs with T4 polynucleotide kinase (T4-PNK), which phosphorylates the 5′ hydroxyl groups and removes the 3′ phosphoryl groups from the RNAs [28,29]. Here, we aimed to determine the added value of phospho-RNA-seq [29] in sRNA abundance and diversity in the EVs and extravesicular plasma subfractions.

## 2. Results

### 2.1. Study Design and General Characterization of sRNA Profiles

To analyze the effects of phospho-RNA-seq in different plasma compartments we obtained the total plasma of three different donors (identified as donors 1, 2, and 3, respectively) and isolated EVs (EV-1, EV-2, and EV-3) and protein-enriched (P-1, P-2, and P-3) fractions using size-exclusion chromatography (SEC) (Figure 1A). Tetraspanins-positive fractions analyzed with flow cytometry and showing low protein concentrations as measured using Nanodrop were pooled and considered as EV-enriched (Figure 1A). Among the three analyzed EV markers, CD9 displayed the highest mean fluorescence intensity (MFI) (Figure 1B), which is in agreement with a recent study which described that serum EVs exhibit higher CD9 expression levels in comparison with other markers [30]. Nanoparticle tracking analysis (NTA) estimated a concentration of 3.04 E + 08 ± 7.54 E + 07 particles/mL with the particles sizes being between 40–120 nm (Figure 1C), which is in agreement with the expected size of circulating EVs [31,32,33]. Vesicular integrity was further determined using cryo-transmission electron microscopy (cryo-TEM) (Figure 1D), and round vesicles with a lipid bilayer were optimally captured in line with other studies [32,34]. Furthermore, classical EV markers, such as Alix, Syntenin, and Flotilin-1 were detected in the EV-enriched fraction through Western blot analysis, while no signal was observed for Calnexin, an endoplasmic reticulum-enriched protein which was used as a negative control (Figure 1E). In parallel, non-vesicular SEC fractions showing the highest protein concentration were pooled as the protein-enriched fraction (Figure 1A). The lack of EVs in this plasma subfraction was confirmed using cryo-TEM (Figure 1D, right panel) and Western blotting, showing no detection of specific EV markers (Figure 1E).

RNA was purified from each plasma and the corresponding EV- and protein-enriched SEC-fractions were treated in parallel with T4-PNK or a buffer (no-T4-PNK) [29]. Afterwards, libraries were prepared and sequenced. In data preprocessing, the 3′ adapter was trimmed in most of the sequences (>90%), indicating that the insert size was essentially less than 42 nt (Appendix A). For annotation, we used the reads that were above 17 nt and less than 42 nt in length (accounting for approximately 50% of the total reads), respectively. The high proportion of too-short reads has been previously observed by Turchinovich and colleagues [35] and Akat et al. [28] and is likely to be indicative of the substantial degradation of unprotected RNA in the blood.

ExceRpt [36], SeqCluster [37,38], and SeqBuster [39] bioinformatic tools were used for sRNA quantification and annotation (Figure 1A). ExceRpt was designed for the analysis of extracellular sRNA using a specific alignment and quantification engine to map and quantify a range of sRNAs without including the rRNAs. Classification based on biotypes showed that under standard conditions the majority of sRNAs in the total plasma were annotated as tRNA (30%), miscRNAs (25%), miRNAs (20%), and protein-coding genes (11%), respectively (Appendix A). Although tRNA and protein-coding gene fragments dominated the sRNA landscape in the EVs and non-EV plasma compartments, the distribution of other sRNA biotypes varied slightly in these compartments (Figure 1F). sRNA mapping onto tRNA fragments (tRFs) were particularly abundant in the plasma EVs, in agreement with previous reports [16], and protein-coding mRNA fragments were especially enriched in the non-EV protein-enriched fractions (Figure 1F). The proportion of miRNA reads within each sample was around 4%, indicating that miRNAs do not dominate the circulating sRNA transcriptome.

### 2.2. Phospho-sRNA Seq Stresses Specific sRNA Profiles in the EV and Non-EV Plasma Compartments

In the non-treated samples, the proportion of sequences less than 17 nt was similar to those of 17–42 nt. In contrast, T4-PNK treatment specifically favored the proportion of sequences of 17–42 nt from 12 to 17% (*p* < 0.0001, chi-squared test), respectively, thus increasing the pool for alignment and annotation (Appendix A). According to ExceRpt, the number of mapped entities with at least one count increased from a mean of 9609 to 38,326 in the EV fractions, and from 16,194 to 40,654 in the protein-enriched fractions upon T4-PNK treatment, respectively. Overall, these data point to an increased diversity of the circulating transcriptome upon phospho-RNA-seq in both plasma compartments.

sRNA distribution based on biotypes (Figure 1F) suggests a variable susceptibility of the sRNA species to phospho-RNA-seq. For example, T4-PNK treatment resulted in a dramatic drop in miRNA detection, highlighting other types of sRNAs that have largely been overlooked under the standard library prep protocols, as previously described [28,29,40]. Decreased coverage of miRNAs when applying phospho-RNA-seq is explained by the increased compatibility of many other sequences with the library prep method. Hierarchical clustering analysis differentiated the T4-PNK-treated RNAs from the non-treated RNAs (Figure 1G and Appendix A), confirming the strong effects of phospho-RNA-seq in sRNA profiles. Specifically, T4-PNK exhibited a strong effect on the protein-coding transcripts in both the EV- and protein-enriched fractions by significantly increasing their number and abundance (Figure 1F). These results are in line with the increased detection of gene fragments in the total plasma RNAs that were submitted to T4-PNK treatment [28,29], which was also observed in the present study (Appendix A).

The proportion of sRNAs mapping onto protein-coding transcripts increased from 19% to 29% in the EVs, and from 32.6% to 53% in the protein-enriched fraction upon T4-PNK treatment (Figure 1F). We captured an average of 4014 and 12,106 unique mRNAs fragments with at least one count in the non-treated and T4-PNK-treated EVs, respectively; and 6262 and 13,536 in the protein-enriched fraction of the non-treated and treated with T4-PNK, respectively. Thus T4-PNK increased the diversity of mRNA fragments by approximately 3-fold in EVs and 2-fold in the extravesicular, protein-enriched fractions, respectively. These data suggest a boosted detection of protein-coding RNA fragments in the EVs, improving the coverage of the precursor mRNA from where they derive, which holds great potential for RNA-based biomarkers discovery, since, unlike miRNAs, mRNAs show an exquisite tissue-specific expression [28,41,42].

Differential expression (DE) analysis confirmed that T4-PNK treatment favors the detection of additional types of sRNAs (Figure 1H). Out of the 1220 and 828 significantly altered sRNAs in the EV- and protein-enriched fractions, 951 and 567 of these were over-represented in the T4-PNK treated samples, respectively, with most of them corresponding to fragments of the protein-coding transcripts (Appendix A). Although sRNAs mapping onto mRNAs are more diverse and abundant in the protein-enriched fraction, DE analysis revealed more significantly over-represented species in the EVs (Figure 1H and Appendix A), suggesting that EVs contain an increased number of mRNA-derived sRNAs with a consistent susceptibility to T4-PNK. Furthermore, hierarchical clustering analysis with the ExceRpt output could optimally separate the EV- and protein-enriched fractions into two groups when using the count matrix generated by phospho-RNA-seq (Figure 1I). These data suggest that T4-PNK treatment stresses the differences between these plasma sub-fractions.

In the data pre-processing step, ExceRpt filters out the sRNAs mapping onto rRNAs; however, these species are highly abundant in the plasma samples [28], and recent studies have indicated that specific rRNAs fragments are uniquely produced [43] and can be bioactive molecules [40]. In addition, ExceRpt classifies tRFs based on the type of isoacceptor; that is, all sRNAs mapping onto tRNAs that carry a specific amino acid are summed up in the same tRNA ID. This type of classification is an oversimplification of the tRFs’ landscape that has been revealed as highly complex. Diverse types of tRFs mapping onto different parts of the mature tRNA are differently expressed under different physiological and pathological conditions, and the study of tRFs expression dynamics and their potential as biomarkers are an expanding research field [44].

To obtain a more in-depth analysis of the rRNA fragments and tRFs, we additionally processed the sequencing data with the SeqCluster tool that detects units of sRNAs (clusters) using a heuristic iterative algorithm to deal with multi-mapped events [37,38]. SeqCluster enables the annotation and quantification of all classes of sRNAs by organizing the transcriptome into groups of sequences that consistently and non-redundantly map onto the same genomic site. Clusters are classified according to the type of precursor where the sRNAs map to, including rRNA and tRNA clusters. SeqCluster revealed that rRNA was the most represented type in the EV- and protein-fractions, which is in agreement with previous results [28,45], followed by gene fragments (gene clusters), tRFs (tRNA-clusters), and sRNAs mapping onto repeat elements (such as the LTR- and SINE-clusters, Appendix A). T4-PNK treatment resulted in an increased detection of rRNA fragments, which is consistent with previous findings using PANDORA-Seq which involves the application of a combined enzymatic treatment with T4-PNK and demethylase to highlight the diversity of the sRNA transcriptome [40].

sRNA biotype distribution (Figure 1F and Appendix A) and the DE analysis showed that tRFs were under-represented upon T4-PNK treatment, especially in the protein-enriched fractions (Figure 1F and Appendix A), where 89% of the DE tRNA clusters were under-represented, compared with a 61% of significant decreased tRFs in the EVs. The differential sensitivity to T4-PNK treatment in the general tRF profiles may be explained by the features of specific fragments that are unequally distributed in the EV- and protein-enriched fractions. For instance, T4-PNK highlighted longer 5′-tRFs derived from the mature tRNA-Leu-TAG-1 and 3′-tRFs derived from tRNA-Arg-GCT-4-1 in EVs compared with the protein-enriched fractions, while these differences were not detected in the non-treated samples (Figure 1J,K and Appendix A). These results suggest that T4-PNK underscores the complexity of tRFs in each plasma compartment. In line with this idea and confirming the ExceRpt analysis (Figure 1I), hierarchical clustering analysis using the sRNA-SeqCluster count matrix only separated the EV- from protein-enriched fractions into two differentiated groups when using expression profiles derived from phospho-RNA-seq (Appendix A).

Overall, these data suggest that phospho-RNA-seq highlights the particularities of the sRNA profiles in different plasma compartments, offers complementary sources of information, and refines the possible identification of the RNA-based biomarkers.

### 2.3. T4-PNK Highlights Different isomiRs Profiles in the EV and Non-EV Plasma Compartments

As previously shown, T4-PNK treatment resulted in a lower read coverage of miRNA in the EVs and extravesicular plasma fractions (Figure 1F and Figure 2A). However, the total number of detected miRNAs (miRNAs with at least one count) with SeqBuster [39], a tool designed to specifically annotate and characterize miRNAs using miRBase as the reference database [46], was not substantially altered by T4-PNK (Figure 2B) (*p* < 0.05, Wilcoxon-matched paired test). Furthermore, specific miRNAs were exclusively captured with phospho-RNA-seq (Figure 2C), which is in agreement with previous findings [29]. These data suggest the existence of specific miRNA configurations that are particularly compatible with phospho-RNA-seq. The separation of the EV- and protein-enriched fractions was achieved in both the T4-PNK-treated and non-treated samples (Figure 2D), indicating that miRNAs are strongly specific to the plasma subfraction, irrespective of phospho-RNA-seq. In non-treated samples, DE analysis revealed a total of eleven miRNAs that were significantly altered between these fractions, with seven being overrepresented in the EV- versus protein-enriched fractions. Upon T4-PNK treatment, nine miRNAs appeared as enriched in the EVs, with only three being common with the non-treated samples (Appendix A). Overall, these data suggest that specific miRNAs are consistently over-represented in the EVs, with T4-PNK treatment stressing the increased expression of additional species in this fraction.

sRNA sequencing analysis has re-defined the miRNA landscape, revealing multiple variants (isomiRs) which differ with respect to the reference, canonical, sequences [47]. IsomiRs mainly originate via imprecise and alternative cleavage during the pre-miRNA processing (5′- or 3′-isomiRs) and post-transcriptional modifications, including 3′-nucleotide additions (3′-add-isomiRs) and nucleotide substitutions (ns-isomiRs) [48]. IsomiRs not only influence miRNA stability and their silencing activity [48,49], but also hold potential to finely classify disease states [50,51,52]. IsomiRs have emerged as a source of biomarkers, with expression patterns specific to tissue types and physiological status, population origin, individual’s gender, and race [53]. Although miRNAs are a low abundant species in both plasma subfractions, we have analyzed the influence of T4-PNK treatment in isomiRs profiles in each plasma fraction using the SeqBuster tool [39], which is specifically designed to characterize isomiRs profiles.

In both plasma subfractions, 3′-isomiRs were highly abundant under standard library prep conditions (Figure 2E). Reference miRNAs were more abundant in EVs compared with protein-enriched fractions, indicating that the canonical forms of these species were specifically loaded in EVs. However, the abundance of the 5′-isomiRs was strongly increased under T4-PNK treatment, especially in the protein-enriched fraction. These results indicate the existence of isomiRs with a 5′-OH poorly compatible with the standard library prep protocols, that only became accessible to adapter ligation after T4-PNK treatment. On the contrary, in the protein-enriched fraction, standard library prep conditions favored the detection of the 3′-isomiRs compared with T4-PNK-treatment, suggesting that miRNA 3′-ends are mostly in the -OH configuration with little sensibility to T4-PNK compared with the miRNA 5′-ends (Figure 2E, lower panel).

In EVs, the sensitivity of the 5′- and 3′-isomirs to T4-PNK was more variable (Figure 2E, upper panel) compared with the consistent over-representation of the 5′-isomiRs and decreased abundance of the 3′-isomiRs in T4-PNK-treated protein-enriched fractions (Figure 2E, lower panel). These observations, along with the general increased detection of the reference miRNAs in the EVs with independence of the treatment, are suggestive of a differential distribution of the types of isomiRs in the different plasma fractions. The favored loading of the reference miRNAs with 5‘-P and a 3′-OH in EVs is compatible with a decreased sensibility to T4-PNK treatment and the subsequent NEBNext library prep. In addition, a variable susceptibility of the 5′- and 3′-ends to phospho-RNA-seq was also detected, suggesting that EVs contain isomiRs with multiple end configurations. This was reflected in the hierarchical clustering analysis using the miRNA count matrix, which showed a lack of separation between the T4-PNK-treated and non-treated samples in the EVs (Figure 2F). On the contrary, the clear differentiation between the T4-PNK-treated and non-treated samples in the protein-enriched fraction reflects the consistent sensibility of the 3′- and 5′-isomiRs to phospho-RNA-seq in the extravesicular plasma compartment (Figure 2F).

### 2.4. T4-PNK Increases the Proportion of Tissue-Expressed Gene Fragments in EVs

Current efforts that are being made to highlight the biomarker potential of circulating RNAs are focused on the identification of the sources (tissues) contributing to the plasma sRNA pool [28,54]. To better understand the effects of T4-PNK in the identification of these sources we used an mRNA-seq tissue atlas comprising major human cells and tissues [28]. This tissue atlas provides a tissue specificity score (TSS) [55], with values above three denoting mRNAs restricted to only a few tissues or cell types.

To compare the sRNA profiles with the tissue atlas, we considered mRNA-derived sRNAs (mRNA-sRNAs) with at least five reads in each sample [54] and took the union of all that were identified in each type of sample (EV- and protein-enriched fractions, treated or non-treated with T4-PNK). In the non-treated samples, we captured 354 and 433 mRNAs-sRNAs in the EV- and protein-enriched fractions, respectively, with 38 presenting a TSS > 3 in each plasma fraction. In the T4-PNK-treated EV- and protein-enriched fractions, we captured a total of 920 and 484 mRNA-sRNAs, with 87 and 45 of these showing a TSS > 3, respectively. Comparing the mRNA-sRNAs with a TSS > 3 between these plasma subfractions, we found that 55 and 13 mRNA-sRNAs were specific to the EV- and protein-enriched fractions, respectively, in T4-PNK-treated samples; and 15 were specifically found in each plasma subfraction in the non-T4-PNK treated samples. These results suggest a significant enrichment of these tissue-specific mRNAs in EVs when sRNAs were treated with T4-PNK (*p* < 0.05, Chi-squared test). The 55 EV-enriched mRNA-sRNAs with a TSS > 3 were enriched in 14 tissues/cells, with the brain, testis, muscle, and blood cells being the more represented sources (Figure 3A). Overall, these data indicate that T4-PNK stresses tissue-specific mRNA fragments in both plasma compartments, favoring this effect in the EVs.

We then compared the 1000 highest-expressed mRNAs for each tissue in the atlas to the mRNA-sRNAs found in the EV- and protein-enriched fractions treated and not treated with PNK. In samples treated with T4-PNK, we found a higher fraction of the top 1000 expressed mRNAs in the EVs compared with the protein-enriched fractions in 17 out of 26 tissues/cells (*p* < 0.05, chi-squared test) (Figure 3B). In the non-treated samples, we did not detect any significantly increased proportions of the top expressed mRNAs in the EVs in any tissue or cell. These data confirm that phospho-RNA-seq specifically highlights the mRNA fragments present in the EVs.

## 3. Discussion

Changes in functional states are accompanied with modifications in the RNA expression patterns, including small regulatory RNAs (sRNAs), which accommodate cell responses to environmental physiological or pathological stimuli. Since sRNAs can be detected in the peripheral biofluids, there is a growing interest in the study of these circulating species as highly sensitive biomarkers. Phospho-RNA-seq has been developed to highlight the diversity of the sRNA transcriptome, providing opportunities to uncover novel informative species. Here, we have applied phospho-RNA-seq to plasma EVs and extravesicular protein-enriched fractions and focused on the 18–42 sRNA fraction that concentrates many of the known functional species. Clustering analysis indicates that T4-PNK stressed the differences between the sRNA profiles in these two plasma compartments, providing a scenario to highlight the specific features of sRNAs.

Standard RNA-seq methods rely on the 5′-P/3′-OH ends of RNAs, such as miRNAs and many tRFs, and RNAs showing alternative ends cannot be properly captured [56]. Independent of T4-PNK treatment, our data show that miRNAs are more abundant in the EVs compared with the protein-enriched fractions. However, miRNAs are little represented species compared with the other types of sRNAs in both the EVs and extravesicular compartments. Even though miRNAs are the most heavily studied and strongly characterized extracellular sRNAs, the present results suggest that other sRNA biotypes, which have been less studied, have potential in RNA-based biomarker discovery.

Phospho-RNA-seq revealed a heightened detection of sRNAs mapping onto protein-coding genes (mRNAs) and rRNAs, which is consistent with previous findings [40]. The increased detection of sRNAs after T4-PNK treatment may encompass a substantial proportion of randomly degraded RNAs with 5′-OH and 3′-P ends. Consequently, mRNA fragments may not necessarily represent the fully processed and functional RNAs. Instead, they can serve as indicators of the relative abundance of precursor mRNA molecules, regardless of whether they are randomly generated or produced through the regulated activity of specific endonucleases. The increased coverage of the mRNA fragments achieved through T4-PNK treatment holds particular significance for RNA-based biomarker discovery [40]. Consistent with this notion, our findings indicate that T4-PNK treatment enhances the presence of gene fragments originating from tissues in the EVs, highlighting their diagnostic potential across diverse biological contexts.

In addition, T4-PNK treatment has the potential to reveal a subset of sRNAs with end configurations that are generated in vivo through regulated mechanisms unrelated to random degradation. RNA metabolism involves enzymatic cleavage that generates OH- or P- in the 5′- and 3′ -ends, or a cyclic phosphate that results from a phosphate bridge between the 2′- and 3′- positions of the pentose (cP) [55]. Multiple enzymes, such as the tRNA splicing endonuclease [57], IRE1 [58], and the ribonucleases T2 [58] and L [59] create cleavage products containing 5′-OH and cP. The demonstrated substrate specificity of several of these endonucleases highlights their context-dependent roles [60,61,62]. For instance, IRE1α is an ER kinase endonuclease that generates 5’-OH ends in certain miRNAs during stress response, terminating their biogenesis and regulating the translation of proapoptotic proteins [60]. Non-random generation of a 5′-OH can also be produced by angiogenin or RNAse T2, which cleave specific tRNAs under stress conditions, with their derived fragments having either protective or toxic roles [63]. In addition, the presence of a 5′-P can be selectively regulated in specific miRNAs to modulate their silencing activity. In cells, a pool of mature inactive miR-34 lacking the 5′-P exists. A DNA-damaging stimulus results in miR-34 5′-P in an ATM- and Clp1-dependent manner, enabling loading into Ago2, and activating its silencing activity [64].

By conducting a paired comparison between the T4-PNK-treated and non-treated samples, our data suggest that specific characteristics of sRNAs can be highlighted under different biological conditions and disease states. For instance, in our study, the paired comparison of the T4-PNK-treated and non-treated samples suggests that miRNAs have differential 5′- and 3′-ends in the EVs and extravesicular fractions. Endogenously, diverse RBPs facilitate miRNA loading into EVs and other RBPs retain miRNAs in the cytoplasm, negatively regulating their sorting into EVs [56]. The Ago2-RISC complex containing functional miRNAs has been proposed to participate in miRNA loading into the EVs, which could explain the enrichment of the canonical miRNAs observed with 5′-P/3′-OH ends in this compartment [12]. However, the 5′- and 3′-isomiRs showed a variable susceptibility to T4-PNK in the EVs, suggesting that other mechanisms may influence the sorting of miRNAs into the EVs. In line with this idea, specific sequence motifs are enriched in the EV RNA content and have been proposed to modulate direct RNA interactions with the lipid membranes during biogenesis, thus controlling their loading into the EVs [65]. In the extravesicular fraction, the obvious detection of the 5′-isomiRs upon T4-PNK treatment suggest that 5′-isomiRs are more abundant than expected, highlighting species produced by non-canonical pre-miRNA processing or by 5′-dephosphorylation of mature miRNAs [60].

The different properties of sRNAs are also stressed by T4-PNK when evaluating tRFs, which have been pointed out as a novel class of biomarkers. Variable susceptibility of tRFs to phospho-RNA-seq has been shown across different tissues and cells [40]. For instance, T4-PNK increased the proportion of tRFs in the brain and liver, and the opposite effect was observed in sperm and HeLa cells [40]. Furthermore, the increased production of specific tRFs is detected under stress conditions in cancer and neurodegenerative conditions, influencing multiple biological processes [66]; overall indicating tissue-, cell-, and context-specific mechanisms in tRF biogenesis. The differential sensitivity to T4-PNK treatment in the EV- and protein-enriched fractions shown here highlights the features of tRFs unequally distributed in these plasma compartments (Figure 1I and Appendix A).

In recent years, there have been substantial scientific attempts to comprehend the RNA composition of EVs since: (1) vesicles act as intercellular communication agents transferring biological information, including RNA between cells [67,68]; (2) EVs can offer direct information from any cell and tissue from where they derive [69,70,71,72]; (3) EVs protect RNAs from degradation in the extracellular environment [69,73]; and (4) they can be isolated from diverse bodily fluids, including plasma, as minimally invasive biomarker sources [74]. Favoring EVs as a meaningful source of RNA biomarkers, we observed that T4-PNK treatment increased the presence of tissue-enriched mRNA fragments in the EVs compared to the extravesicular fractions. Additionally, T4-PNK treatment improved the detection of significantly altered sRNAs in the EVs, indicating a more consistent sRNA landscape. Although our data point to the benefits of T4-PNK treatment in RNA biomarker discovery, several limitations need to be acknowledged. Phospho-RNA-seq does not address the incompatibility of the traditional RNA sequencing methods in detecting methylated sRNAs [40] and other yet-to-be-identified problematic modifications. Therefore, further developments are necessary to enhance the methods for the comprehensive discovery and identification of meaningful sRNAs. Additionally, specific functional validations will be required to determine the biological relevance of the newly identified species. This validation will help ascertain whether they are genuine sRNAs or by-products that reveal an indirect activity of nucleases within cells. Nevertheless, our data point to the possibility that measuring EV- and extravesicular-derived RNAs after T4-PNK treatment can offer complementary information and could be parallelly implemented under precision medicine settings to finely define a physiological status.

## 4. Materials and Methods

### 4.1. Blood Collection and Sample Processing

Whole blood samples (*n* = 3) were provided by the Banc de Sang i Teixits (BST, Barcelona, Spain). The blood, collected using citrate-based anticoagulant bags (ACD), was certified by the provider as testing negative for HBs Ag, Anti-HCV, Anti-HIV 1+2, syphilis serology, VHB, HCV, and HIV. Cell-free plasma was obtained through sequential centrifugations, as previously described [75,76,77]. Briefly, blood samples were centrifuged for 10 min at 500× *g* and the cellular pellet was discarded. The obtained supernatant was then centrifuged at 2500× *g* for 15 min followed by a final centrifugation step at 16,000× *g* for 15 min.

### 4.2. Extracellular Vesicles Isolation

Two mL of cell-free plasma were used to isolate EVs using size exclusion chromatography (SEC), as previously described [77]. Briefly, Puriflash columns Dry Load Empty 12 g (20/pk) (Interchim, Montluçon, France) were manually loaded with Sepharose™ CL-2B (GE Healthcare, Chicago, IL, USA) and plasma samples were then applied. Sample elution was performed using PBS1x and constant 500 μL fractions were collected. All fractions were characterized for their protein concentration measuring their absorbance at 280 nm using a Nanodrop (NanoDrop One/Onec ThermoFischer Scientific, Waltham, MA, USA). Fractions showing a low/minimal protein concentration were analyzed with bead-based flow cytometry for the presence of three classical vesicular markers: CD63, CD9, and CD81, as previously described [78,79]. Fifty μL of each low-protein SEC fraction were incubated with aldehyde/sulphate-latex beads (4 μm; Invitrogen, Carlsbad, CA, USA) for 15 min and were then resuspended in 1 mL bead-coupling buffer (PBS supplemented with 0.1% BSA and 0.01% NaN_3_) for overnight incubation at room temperature. EV-coated beads were then labeled at 4 °C with anti-CD9 (Clone VJ1/20), anti-CD63 (Clone TEA 3/18), and anti-CD81 (clone M38), which were all purchased from Immunostep (Salamanca, Spain), and FITC-conjugated secondary goat anti-mouse antibody (SouthernBiotech, Birmingham, AL, USA) was used for detection. Samples were analyzed using flow cytometry in a BD SRFortessaSORP + HTS cytometer and FACS Diva Software (version 9.1, BD Biosciences, San Jose, CA, USA).

The four fractions showing the highest mean fluorescence intensity (MFI) fold change were pooled and identified as the EV-enriched fraction. Additionally, the four fractions with the highest protein concentration according to the Nanodrop determinations were pooled as the protein-enriched fraction.

Total protein concentrations from the different pooled fractions were determined using the bicinchoninic acid assay (MicroBCA^TM^ Protein Assay kit, ThermoFisher Scientific, Waltham, MA, USA) following the manufacturer’s instructions.

### 4.3. Quantification and Morphological Characterization of the EVs

Size distribution and an estimated concentration of isolated EVs were determined using nanoparticle tracking analysis (NTA; NanoSight NS300, Malvern Instruments Limited, Worcestershire, UK) following the manufacturer’s instructions. EV samples were diluted at 1:40 with sterile and filtered PBS, and three videos of 60 s at 25.0 frames per second (FPS) at a camera level of 10 were recorded. The mean values for size (nm) and concentration (particles/mL) of the three replicates were considered as the final measurement.

Additionally, EV-enriched fractions were examined using cryo-transmission electron microscopy (cryo-TEM), as previously described [32]. Briefly, vitrified specimens were prepared by placing 3 μL of a sample on a Quantifoil. 1.2/1.3 TEM grid, blotted to a thin film, and plunged into liquidethane-N2(l) in the Leica EM CPC cryo-work station. The grids were analyzed with a Jeol JEM 2011 transmission electron microscope, and images were recorded on a Gatan Ultrascan 2000 cooled charge-coupled device (CCD) camera with the Digital Micrograph software package (version 3.x Ametek, Berwyn, PA, USA).

### 4.4. Western Blot Analysis

EV- and protein-enriched fractions were subjected to Western blotting for the detection of specific EV markers. Equal amounts of protein (25 μg) of each sample measured by the Micro BCA^TM^ Protein Assay Kit (Themo Fisher Scientific, Waltham, MA, USA) was considered.

Samples were then loaded in NuPAGE 4–12% Bis-Tris polyacrylamide gels (Thermo Fisher Scientific, Waltham, MA, USA) and transferred to a nitrocellulose membrane using the iBlot2^®^ Transfer Stack supports (Thermo Fisher Scientific, Waltham, MA, USA). Immunoblots were probed with the EV-markers anti-Alix antibody (1:1000, Proteintech, Rosemont, IL, USA; Cat. No: 12422-1-AP), anti-Flotilin1 (1:1000, BD Biosciences, Franklin Lakes, NJ, USA; Cat. No: 610820), and anti-Syntenin (1:1000 Abcam, Cambridge, UK; Cat. No: 133267). Anti-Calnexin (1:1000, Abcam, Cambridge, UK; Cat. No: 22595) was used as a negative control. Membranes were incubated with the primary antibodies overnight at 4 °C under conditions of agitation. After washing them, membranes were incubated with the secondary antibody anti-rabbit HRP-conjugated IgG (1:5000, Thermo Fisher Scientific, Waltham, MA, USA; Cat. No: 31460) or with the anti-mouse HRP-conjugated IgG (1:5000, Thermo Fisher Scientific, Waltham, MA, USA; Cat. No: 31430). Immunoreactive bands were visualized using the Western blotting SuperSignal^TM^ West Pico PLUS Chemiluminescent Substrate Reagent (Thermo Fisher Scientific, Waltham, MA, USA) and images were acquired using the Chemidoc^TM^ Imager (Bio-rad, Hercules, CA, USA).

### 4.5. Library Preparation and Sequencing

EV-enriched pools were further concentrated with the Amicon Ultra-2 100 kDa (Merck Millipore, Darmstadt, Germany) following the manufacturer’s protocol before RNA isolation. Protein-enriched fractions were concentrated 10-fold by the speed vacuum at room temperature (RT) (SC210ASpeedVacPlus, ThermoSavant). RNAs from the total plasma, EV-, and protein-enriched fractions were isolated using the miRNeasy Serum/Plasma kit (Qiagen, Venlo, The Netherlands) according to the manufacturer’s protocol, and were further treated with T4-PNK (New England Biolabs, Ipswich, MA, USA) as previously described [29], and following the scheme displayed in Figure 1A. Briefly, half of the fractions total volume was added to 1/7 volume of the T4 buffer, adenosine 5′-triphosphate (ATP), and T4-PNK, and was incubated at 37 °C for 30 min and at 70 °C for 5 min. RNA samples were then washed using the RNA Clean-up and concentrator Kit (Zymo Research, Irvine, CA, USA) following the manufacturer’s protocol. Samples lacking T4-PNK treatment were also submitted to the additional clean-up step in parallel. Isolated RNA was further precipitated with glycogen (20 μg/μL), 10% 3 M AcNa (pH 4.8), and 2.5x volume of 100% ethanol. sRNA libraries were prepared using the NEBNext Small RNA Library Prep Set for Illumina (New England Biolabs, Ipswich, MA, USA), according to the manufacturer’s instructions. Individual libraries were subjected to quality analysis using a Bioanalyzer 2100 (Agilent, Santa Clara, CA, USA), and size selection (140–165 bp) was performed in 6% polyacrylamide gel. Indexed libraries were equimolarly pooled and sequenced in an HiSeq2500 (Illumina) in single-end read modes and 1 × 50 cycles at the Centre of Genomic Regulation (CRG, Barcelona, Spain).

### 4.6. Bioinformatic Analysis for sRNA Characterization

The quality of the sequenced fastq files was checked using FastQC software (v0.12.1) (http://www.bioinformatics.babraham.ac.uk/projects/fastqc/ accessed on 17 December 2020). Adapter trimming was performed with Cutadapt [80], and the trimmed reads were aligned to the human genome (version Ensemble hg 19) using the STAR aligner (version 2.7.9a) [81]. Quantification and annotation were performed with the ExceRpt [36], SeqCluster [37], and Seqbuster tools [39].

### 4.7. Data Analysis and Statistical Testing

For downstream analyses, we considered RNAs with 10 or more reads in at least three samples. Differential detection of the sRNAs between the T4-PNK-treated and non-treated samples was performed using the negative binomial linear models implemented within the DESEq2 R/Bioconductor (v4.0.0.) package [82]. To take into account the paired nature of the design, the donor identifier was included as a covariate in the models.

The chi-squared test was used to evaluate the tissue-enriched proportions of the protein coding fragments. A Benjamini–Hochbergh correction was used to correct the *p*-values for multiple testing. A 5% statistical significance level was considered throughout all the analyses.

## 5. Conclusions

In summary, our results confirm that the use of phospho-RNA-seq can provide a more comprehensive/extended view of the plasma sRNA transcriptome, stressing the difference between the plasma compartments and highlighting the novel species and potential RNA-based biomarkers. Our data highlight that the study of sRNAs in plasma EVs and extravesicular compartments after T4-PNK treatment can offer complementary information and may be useful in precision medicine settings to finely define the physiological and pathological states. The present findings suggest that alternative sRNA biotypes beyond miRNAs have potential in RNA-based biomarker discovery, and that EVs may offer a more informative source of the meaningful biomarkers.

## Figures and Tables

**Figure 1 ijms-24-11653-f001:**
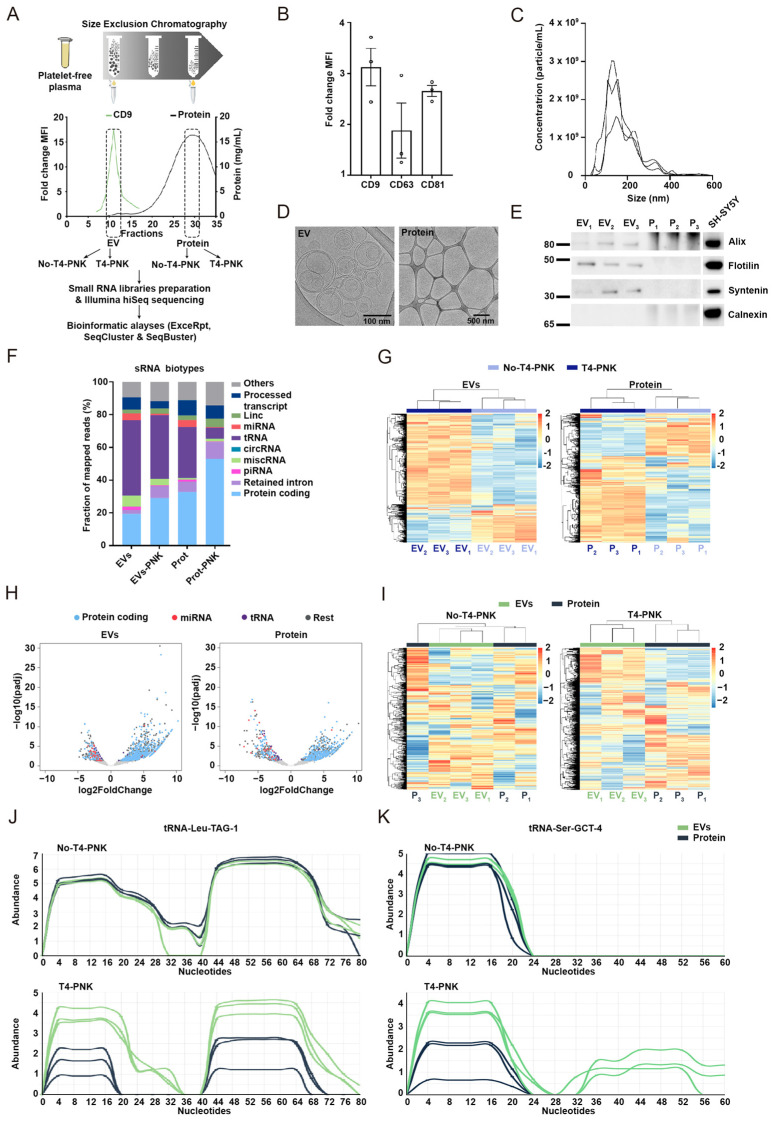
Specific small RNA profiles are found in the extracellular vesicles and protein-enriched fractions upon T4-PNK treatment. (**A**) Schematic illustration of the workflow followed in this study. (**B**) Fold change in the MFI values for the EV markers CD9, CD63, and CD81 in the EV-enriched fraction pooled after SEC. MFI fold changes were calculated as referenced to the negative control. (**C**) NTA size distribution profiles of the pooled EV-enriched fractions for the three different samples. (**D**) Representative cryo-TEM image of the EV- and protein-enriched fractions obtained after plasma SEC subfractionation. (**E**) Evaluation of EV markers by Western blot. SH-SY5Y lysate was included in the analysis as the positive control. (**F**) Fraction of short RNA reads mapping to distinct classes of small RNAs as annotated using the ExceRpt tool for the EV- and protein-enriched fractions, both after treatment with T4-PNK and without T4-PNK treatment. (**G**) Heatmap of hierarchical clustering analysis of sRNA profiles in EV- and protein- enriched pools using the ExceRpt tool. (**H**) Volcano plot showing differentially expressed sRNAs in T4-PNK treated samples versus non-treated samples, both in the EV- and protein-enriched fractions as annotated using ExceRpt. Colored dots represent the significantly deregulated sRNAs (|log2FoldChange| > 0.58, adjusted *p* < 0.05). miRNAs, tRNA, and gene fragments are specifically highlighted. (**I**) Heatmap of hierarchical clustering analysis of sRNAs identified in the No-T4-PNK treated and T4-PNK-treated samples. (**J**) Normalized abundance of sequences belonging to tRNA-Leu-TAG-1 (SeqCluster tool) from the EV- and protein-enriched fractions, both treated and not treated with T4-PNK. (**K**) Normalized abundance of sequence mapping onto tRNA-Ser-GTC-4 (SeqCluster tool) from the EV- and protein-enriched fractions, both treated and not treated with T4-PNK. Data in (**B**) are represented as mean ± SEM (*n* = 3 samples per group).

**Figure 2 ijms-24-11653-f002:**
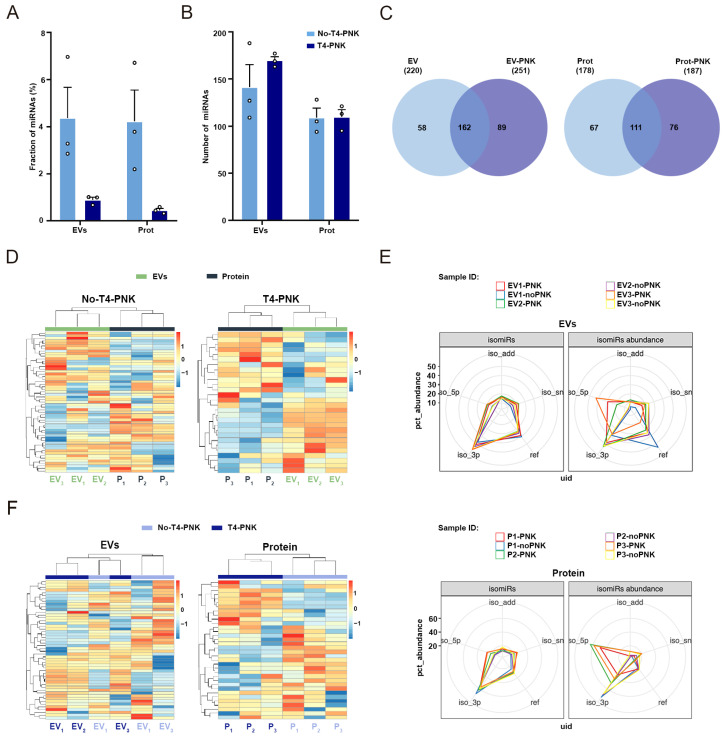
Phospho-RNA-Seq stresses the detection of specific miRNAs in the EV- and protein-enriched plasma fractions. (**A**) Fraction of reads that align to miRNAs found in the T4-PNK-treated and non-treated samples using the ExceRpt tool. (**B**) Absolute number of miRNAs identified using ExceRpt in the EV- and protein-enriched fractions treated and not treated with T4-PNK. (**C**) Venn diagrams of the miRNAs identified by ExceRpt in both the treated and non-treated EV- (left panel) and protein-enriched fractions (right panel). miRNAs identified with >0 RPM in each sample were considered. (**D**) Heatmap hierarchical clustering analysis of the miRNAs identified using the SeqBuster tool in the non-treated and treated samples. (**E**) Radar plots showing the number of the different types of isomiRs (left plots) and the abundance of each type of isomiR (right plots) in the EV- and protein-enriched fractions as analyzed with SeqBuster. Each type of isomiR is labelled as either Iso 5p (miRNAs that vary in the 5′ end); Iso 3p (miRNAs that vary in the 3′ end); Iso add (miRNAs varying in the 3′end due to nucleotide additions); and Iso sn (miRNAs that vary because of a nucleotide substitution), respectively. Ref indicates the canonical (most abundant) reference miRNA. (**F**) Hierarchical clustering analysis of the miRNAs identified using the SeqBuster tool in the EV- and protein-enriched fractions. Data in (**A**,**B**) are represented as mean ± SEM (*n* = 3 per group).

**Figure 3 ijms-24-11653-f003:**
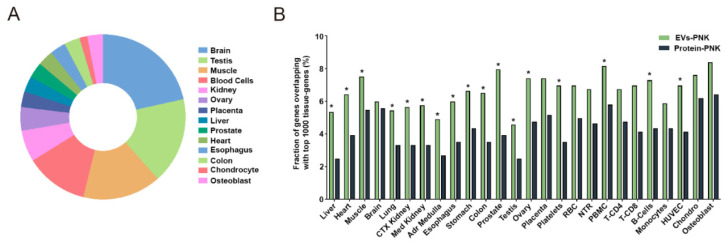
T4-PNK treatment in EVs highlights tissue-specific mRNA fragments. (**A**) Genes with a TSS > 3 overlapping with sRNAs mapped onto protein coding (with >5 reads in each of the samples) specifically found only in the EV-enriched fractions treated with T4-PNK. (**B**) Percentage of genes expressed (sRNAs that mapped onto protein coding with >5 reads in each of the samples) in the EV- and protein-enriched fractions treated with T4-PNK that overlap with the top 1000 expressed genes in each tissue. The dataset from the ExceRpt tool was used (*n* = 3 per group). The chi-squared test was used to evaluate whether T4-PNK treatment resulted in significant differences in the proportions of tissue-enriched protein coding gene fragments between the EV- and protein-enriched fractions (*p* * < 0.05).

## Data Availability

Raw data from sequencing experiments are available upon request. We have also submitted all relevant data regarding our protocols and procedures experiments to the EV-TRACK knowledgebase (EV-TRACK ID: EV230052).

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
