# Peer review of "Phospho-RNA-Seq Highlights Specific Small RNA Profiles in Plasma Extracellular Vesicles"

_ijms, 2023, doi:10.3390/ijms241411653_

Round 1

Reviewer 1 Report

The manuscript is focused on the differences between two sRNA cloning protocols which have been applied to a cohort of 9 samples: 3 EVs, 3 plasma and 3 RNA extractions from the protein enriched fraction of plasma.

Major point: Based on my experience, phospho RNA-seq enhances the chance of recovering random RNA fragments rather than whole RNA molecules.
The authors argue that standard protocol for sRNA-seq will exclude RNA molecules without a phosphate group at 5' end, as well as those carrying phosphates at 3' end. This statement is in fact correct, however, to better explain the possible significance of the phospho-sRNA-seq, the authors should provide convincing evidence that such (i.e. 5'OH, 3'phospho) sRNA molecules (other than random RNA debris) actually exist. This includes identification (by the authors or in the literature) of enzymes which yield 5'OH/3'phospho RNA molecules (apart from RNA degradation pathways). If no explanation for such unusual 5'/3' ends other than random fragmentation can be supported by the data or the literature, then the overall significance of their findings, in my opinion, will not suffice to grant publication on IJMS. Indeed, there are several reports (actually, all those upon which the current sRNA library preparation protocols are based) reporting that random RNA degradation fragments do not carry 5'phosphate groups, hence omitting T4PNK is a good strategy to deplete those undesired degradation products from libraries.
In fact, the major take home message of this manuscript in its current form is that sRNAs arising from long RNA molecules (not previously reported to yield functional sRNAs) are enriched in phospho-sRNA-seq (which in the absence of other data I would interpret as degradation fragments) while previously reported sRNAs associated with a putative function (miRNAs, tRNA fragments) are depleted with the same protocol.
This equals to the current, well established and  widespread view (supported by countless papers) that omitting T4PNK step ensures the appropriate enrichment of functional sRNAs over random fragments in sRNA-seq protocols.

For this major issue, I would recommend rejection at this stage, and possible resubmission of a completely reformatted manuscript with experiments proving the biological significance (in terms of biogenesis and phenotype) of at least one of the sRNA species which can only be identified through phospho-sRNA-seq over RNA-seq

Minor points:

1) What is the biological significance of the 17 and 42 nt cutoffs?

2) If statistically significant differences were found in histograms, those should be highlighted appropriately, reporting the statistical test used.

3) panel letters (A-I) in figure 1 do not match those (a-j) reported in the figure legend

4) Supplementary Figure 4 suggests that the data reported in the last (I/j) panel of Fig 1 are not representative as for other clusters on other tRNA genes completely different outputs were obtained. The authors should either report in the main figures three profiles representative of the different outcomes, or keep all profiles as supplementary figures, removing the one reported in main Fig 1

Reviewer 2 Report

In their article “Phospho-RNA-Seq highlights specific small RNA profiles inplasma extracellular vesicles“, Solaguren-Beascoa and colleagues compare different strategies to analyze sRNAs in EVs and extravesicular plasma. As a result, the authors favorize the use of phosphor-RNA-seq by T4-PNK treatment of sRNAs to detect otherwise inaccessible RNAs. This method also increased the number of tissue-specific sRNAs in EVs. This method appears to be useful for searching and detecting biomarkers in different diseases.

Author Response

We thank the reviewer for the positive feedback on our work.

Reviewer 3 Report

This manuscript entitled "Phospho-RNA-Seq highlights specific small RNA profiles in plasma extracellular vesicles" by Solaguren-Beascoa M. et al. indicated tissue specificity of sRNA in plasma EVs by Phospho-RNA-Seq. This manuscript is very interesting. But, some corrections may be needed. In the Introduction section, it was better to describe molecular mechanisms of encapusulation of sRNAs into EVs previously reported with references. In fig.1, it was better to analyze the expression of molecular markers of EVs and negative control by western blot. In addition, it was better to analyze the relationship of small and large EVs with Phospho-RNA using other isolation methods of EVs, such as gradient-ultracentrifugation and fractionation. In fig.1D, it was better to analyze by using gold-colloid staining. In the materials and Methods section, it was better to describe the isolation procedure of EVs in more detail. In the Discussion section, it was better to describe molecular mechanisms of formation of Phospho-RNA in EVs. 
